# Review of Botnet Attack Detection in SDN-Enabled IoT Using Machine Learning

**DOI:** 10.3390/s22249837

**Published:** 2022-12-14

**Authors:** Worku Gachena Negera, Friedhelm Schwenker, Taye Girma Debelee, Henock Mulugeta Melaku, Yehualashet Megeresa Ayano

**Affiliations:** 1Addis Ababa Institute of Technology, Addis Ababa University, Addis Ababa 445, Ethiopia; 2Institute of Neural Information, University of Ulm, 89069 Ulm, Germany; 3Ethiopian Artificial Intelligence Institute, Addis Ababa 40782, Ethiopia; 4College of Electrical and Computer Engineering, Addis Ababa Science and Technology University, Addis Ababa 16417, Ethiopia

**Keywords:** botnets, software defined networks, internet of things, machine learning

## Abstract

The orchestration of software-defined networks (SDN) and the internet of things (IoT) has revolutionized the computing fields. These include the broad spectrum of connectivity to sensors and electronic appliances beyond standard computing devices. However, these networks are still vulnerable to botnet attacks such as distributed denial of service, network probing, backdoors, information stealing, and phishing attacks. These attacks can disrupt and sometimes cause irreversible damage to several sectors of the economy. As a result, several machine learning-based solutions have been proposed to improve the real-time detection of botnet attacks in SDN-enabled IoT networks. The aim of this review is to investigate research studies that applied machine learning techniques for deterring botnet attacks in SDN-enabled IoT networks. Initially the first major botnet attacks in SDN-IoT networks have been thoroughly discussed. Secondly a commonly used machine learning techniques for detecting and mitigating botnet attacks in SDN-IoT networks are discussed. Finally, the performance of these machine learning techniques in detecting and mitigating botnet attacks is presented in terms of commonly used machine learning models’ performance metrics. Both classical machine learning (ML) and deep learning (DL) techniques have comparable performance in botnet attack detection. However, the classical ML techniques require extensive feature engineering to achieve optimal features for efficient botnet attack detection. Besides, they fall short of detecting unforeseen botnet attacks. Furthermore, timely detection, real-time monitoring, and adaptability to new types of attacks are still challenging tasks in classical ML techniques. These are mainly because classical machine learning techniques use signatures of the already known malware both in training and after deployment.

## 1. Introduction

Digitization has brought a lot of advantages in a computing context. As a result, nations have increased their dependencies on digitization, software-defined networks (SDN), and the internet of things (IoT) [1,2]. In the digitization era, the major challenge we are facing is cybersecurity. Cyber-attacks can disrupt and sometimes cause irreversible damage to several to the economy. For instance, they can cause an electrical blackout [3], theft of financial institutions [4], failure of military equipment [5], and breaches of national security secrets [6]. Also, they can result in the theft of valuable and sensitive data like medical records. Besides, they can disrupt phone and computer networks or paralyze systems. According to the Outpost24 report [7], more than 3800 big corporations experienced data breaches in 2019 that resulted in dire consequences. Despite all these attacks, there has been a lack of awareness and system security, as well as a lack of quality research findings that can bring an effective solution to contain malware attacks [8].

An IoT network is a network of physical objects or things empowered with limited computation, storage, and communication capabilities as well as embedded with electronics (such as sensors and actuators), software, and network connectivity that enables these objects to collect, sometimes process, and exchange data [9,10].

Several applications and services benefited a lot from IoT, including critical infrastructure management, agriculture, military, smart grid/cities, and personal healthcare [9]. However, IoTs are vulnerable to malware attacks, specifically Botnet malware [11]. The vulnerability is mainly because IoT devices lack sufficient security mechanisms due to a lack of secured credential usage and updates [12], and unsecured communication [13].

### 1.1. IoT Botnets

The term botnet is derived from two words, robot and network. A bot is a program that performs user-centric tasks independently without any interaction from a user [14]. Based on how to control the bots, there are two types of botnet architecture, a centralized client-server and a decentralized peer-to-peer (P2P) approach. In a centralized client-server approach, the botmaster controls and monitors all bots from a single central point through command and control (C&C) commands. In a decentralized mode, each bot acts as a server and client that distributes and receives commands [15].

IoT botnet becomes part of the network where they maliciously control computing devices by performing three operations. The first operation is locating vulnerable devices (scanning). Then, a suitable bot that fits the architecture of the vulnerable device is installed (propagation). Finally, an attack is initiated via command and control operation (attack operation) [16]. For instance, the Mirai botnet contains attack vectors, a scanner process that actively seeks other devices to compromise, and the C&C that controls the compromised devices (bots) to facilitate further propagation and initiate an attack [17].

An IoT botnet consists of a network of devices, such as cameras, routers, DVRs, wearables, and other embedded devices, that are infected with malware [18]. Some of the most common Botnet malware attacks that affect IoT include Mirai and BASHLITE [17]. These botnets have been designed to launch several types of diverse cyber-attacks. The most common attack is identity theft, where the Bot-code infecting a machine gathers sensitive user information and sends it to the botmaster. Similarly, in e-mail spamming attacks, infected devices are used to produce and send fake e-mails. In a key-logging type attack, the user’s input is logged and transferred to the botmaster [16].

### 1.2. Software Defined Networks (SDN)

The vertical integration of data and control planes in traditional networks poses challenges in the installation, management, re-configuration, and maintenance of the network [19,20]. To mitigate these challenges and provide additional capabilities to networks, SDN was introduced in early 2010 [20]. As shown in Figure 1 in an SDN architecture, there is a separation among data, control, and management plane. In the architecture shown in Figure 1, the control plane is decoupled from the network hardware and implemented as software. The SDN controller acts as the brain of the SDN system that manages the flow of control commands to the data plane via southbound application program interfaces (APIs). In addition, it oversees the flow of controls to the management plane via the northbound APIs. Furthermore, the SDN controller facilitates the communication between controllers and legacy networks via the eastbound and westbound APIs, respectively [19].

The management plane ensures seamless network operation and monitors the performance. From the perspective of traditional networks, the management plane handles the management of fault, configuration, accounting, performance, and security of an SDN system [21]. On the other hand, the data plane contains networking devices such as routers and switches. However, these devices do not make autonomous decisions like their equivalents in traditional networks. In contrast, they communicate with the SDN controller through southbound APIs such as OpenFlow to decide flow entries [22].

Generally, SDN has superiority over the traditional network in terms of programmability, flexibility, usability, security, on-demand quality service, topology discovery, user mobility support, and fault localization. Besides, SDN handles interoperability of heterogeneous device across the network and facilitate rapid innovation [19].

### 1.3. SDN-Enabled IoT

According to Li et al. [23], the orchestration of IoT with SDN can reduce the security vulnerability of IoT devices. This reduction can be achieved by provisioning sufficient computational resources during the deployment of complex anti-malware applications. SDN also enhances IoT networks’ scalability, management, resource usage, and resilience against attacks [23]. Furthermore, the SDN decoupled the traditional network architecture by separating the control and data planes. This characteristic has brought simplicity within the network through simplifying re-configuration, control and management, enhancing interoperability, and improving network programmability. The other enhancement that the orchestration brought is; the improvement of the security of the traditional IoT networks [24]. However, the SDN-based IoT networks are still vulnerable to botnet attacks, and several techniques have been proposed to alleviate these attacks [25,26,27,28,29].

An SDN-enabled IoT architecture consists of four hierarchical layers. The lowest layer in the hierarchy is called the perception layer. This layer consists of sensors and actuators that sense an environment and act upon the environment, respectively. The layer that comprises the SDN gateways and routers is called the communication/network layer. This layer, under the control of the SDN controller, plays the role of data forwarding. The third layer is called the computing layer. It consists of an SDN controller and billing mechanisms. The last layer in the hierarchy is the service/application layer, where developers and operators build IoT services through programming the SDN controller [30]. In SDN-enabled IoT, the SDN controller plays a critical role in managing flow control to the switches/routers via southbound APIs and the applications via northbound APIs [31].

### 1.4. Machine Learning in SDN

Unlike the conventional rule-based programming approach, machine learning techniques solve complex problems by learning from the dataset compiled from past experiences. This process is accomplished in a variety of methods. The first method is through generating a model by processing the labeled dataset and performing a prediction or classification for a new instance of input data by executing the trained model. This technique is known as supervised machine learning [32]. The second method is trained by providing a set of unlabelled datasets to the ML algorithm and letting the ML algorithms identify the trends of similarity within the dataset. These techniques are known as unsupervised machine learning [33]. The other learning approach is reinforcement learning, the agent takes actions and continuously senses the environment to update the knowledge base on intelligent policies guided by reward [34]. So, in reinforcement learning, the machine explores the environment and executes an action based on its state and the environment condition to maximize the goal [32]. Machine learning techniques have been used in automating and aiding experts in different application areas such as medical sciences [35,36,37,38], agriculture [39,40,41], cyber security [42,43,44].

In SDN, machine learning techniques have been implemented in the controller of the software-defined network to add intelligence to the controller. The machine learning techniques can enhance the SDN controller efficiency in the prediction of QoS, network traffic classification, resource management, prediction of QoE, and overall security [28].

### 1.5. Paper Organization

The rest of the paper is organized as follows, Section 2 discusses recent related works, and Section 3 elaborates on the technique used to conduct the review and research questions addressed in this review. The major botnet attacks in SDN-IoT networks are investigated and presented in Section 4. The commonly used machine learning techniques for botnet attack detection in SDN-IoT networks are discussed in Section 5. The findings of this review are discussed in Section 6, and a brief discussion on existing challenges and future directions is provided in Section 7. Finally, the conclusion is presented in Section 8.

## 2. Related Work

So far, to the best of our knowledge, literature surveys related to machine learning-based botnet attack detection in SDN-orchestrated IoT networks are very limited both in number and scope. However, there exists literature that investigated the security features that the SDN-IoT orchestration introduces in combating cyber-threats in IoT [45,46,47].

Wang et al. [25] investigated a few works of literature to identify state-of-the-art DDoS attacks and the proposed machine learning mitigation mechanisms. Accordingly, they tried to show machine learning techniques that use features of the incoming packets in deciding whether the traffic is a DDoS attack. Relatively, a more extended survey was provided by Shinan et al. [48]. The authors identified literature that focused on detecting botnets using machine-learning techniques in traditional networks. Besides, they discussed recent studies on machine learning-based botnet detection in an SDN. However, the survey lacks in distinguishing the IoT environment from the traditional network, including the nature of botnet attacks [49]. In addition, the survey did not include machine learning-based botnet attack deterrence mechanisms in SDN-IoT orchestrated networks. Similarly, Restuccia et al. [50] have investigated the uses of machine learning and SDN in establishing secured IoT networks. Again, their survey did not include the vulnerability of the SDN-enabled IoT networks for different types of botnet attacks.

The limitation of SDN-enabled IoT networks in malware attack detection investigated by Pajila and Julie [47]. Though the orchestration of IoT with SDN improves botnet attacks such as DDoS attacks, there are still holes that make the SDN-IoT system vulnerable to a new stream of assault [47]. Pajila and Julie [47] have surveyed and discussed the potential of using machine learning techniques as one DDoS attack detection mechanism in SDN networks. However, very few works of literature were included in their survey. In addition, their survey did not discuss the use of ML techniques for botnet attack prevention.

The Summary of related works is shown in Table 1 with their contribution and limitations. So, our review aimed at filling all the gaps discussed above and providing a comprehensive review of works in the use of machine learning techniques for detecting and mitigating botnet attacks in SDN-enabled IoT networks.

## 3. Methods

This section presents the methodology for reviewing the ML-based techniques in botnet attack detection in SDN-enabled IoT networks. So, the objectives, research questions, search strings for identifying research studies, and selection criteria are discussed in this section.

So, the main objective of this literature review is to investigate existing research studies proposed for deterring botnet attacks in SDN-enabled IoT networks using machine learning techniques. Research questions are formulated to address the core points of the review as shown in Table 2. Search strings and databases used to find literature for this review are indicated in Algorithms 1 and 2, respectively. Based on the inclusion and exclusion criteria shown on Table 3, some of the literature found in listed databases using the indicated search strings are excluded.
**Algorithm 1:** Pseudocode for Defining Search StringSearch_String = [(“DOS” **OR** “DDOS” **OR** “Port Scanning” **OR** “brute forcingattacks” **OR** “credential stuffing”)**AND**(“Attack” **OR** “Malware” **OR** “Network Security for IoT” **OR** “Cyber attack onSDN-IoT”) **OR** “BotNet Attack”)**AND**(“Machine Learning Methods” **OR** “Deep Learning Methods” **OR** “CNN” **OR**“DNN” **OR** “LSTM” **OR** “GRU” **OR**“RNN” **OR** “Classical Machine LearningMethods” **OR** “SVM” **OR** “RF” **OR** “KNN” **OR** “LR” **OR** “DT” **OR** “NB”)**AND**(“Detection” **OR** “Classification” **OR** “Prevention”)]

**Algorithm 2:** Pseudocode for Generating Potential Review Papers


Search Databases←Springer_Link,MDPI,Science_Direct,Wiley_Online,


IEEE_Xplore


**{Initialization:}**
Area_Keyword ←[BotNet,Software_defined_Network,Malware,Attack,IoT]Attack_keywords ←[DOS,network−probing,DDOS,backdoor,Phishing,
credentialstealing]   Method_keywords ←[Deep_Learning_Methods,Classical_Machine_Learning_Methods]   Target_keywords ←[Detection,Classification]Search_String ←″Algorithm1″**for**

attack∈Attack_keywords

**do****    for**

keyword∈Area_keywords

**do****        for**

target∈Target_keywords

**do****            for**

method∈Method_keywords

**do**                Search_String    =Algorithm1**                for **database∈Databases

**do**                    Paper_List1←databases.search(Search_String)
**                end for**

**            end for**

**        end for**

**    end for**

**end for**
Inclusion_Criteria = [IC1,IC2,IC3,IC4,IC5,IC6]Exclusion_Criteria = [EC1,EC2,EC3,EC4]Paper_List2 ←Apply.Inclusion_Critera(Paper_List1)Paper_Final_Lists ←Apply.Inclusion_Critera(Paper_List2)


## 4. Botnet Malware in SDN Orchestrated IoT

SDN-based IoT networks are still vulnerable to Botnet attacks [25,28]. Although there are different botnet attack classification techniques, in SDN-enabled IoT networks bots can be used to generate DDoS attacks [25], information stealing attacks [48], and phishing attacks [55] on the network by getting a foothold in the network through backdoor vulnerability [56] and network-probing attacks [57].

### 4.1. Distributed Denial of Service Attacks

The DDoS attack is one of the most common botnet attacks in SDN-enabled IoT networks. It is an attack that hinders systems availability by consuming systems’ resources. The DDoS attacks need three environmental agents to attack SDN-enabled IoT systems. These are attacker, handler, and target environment [58]. In the attacking phase, the botmaster selects a botnet based on its functionality and structure. Then, new devices with security holes are identified through the botmaster that subsequently infects and controls them. In the handling environment, the botnets use the Internet relay chat (IRC) or HTTP for communication between the attacker and botmaster’s communication and control server to further propagate in the network. On the other hand, the target environment consists of the newly infected devices [58,59].

Based on the IoT’s depleted resources by the attack, the DDoS attack can be categorized into application layer attacks, resource exhaustion attacks, and volumetric attacks [58]. Application layer attacks are the most troublesome type of DDoS attacks to be detected. DDoS attacks on this layer are often mistaken for application instability, and implementation inefficiency [59]. The resource exhaustion DDoS attacks mainly exploit the communication protocols and deplete the computation resources of the network that would have been used for legitimate users. The third type of DDoS attack is a volumetric attack that affects the system by saturating the communication channel. These types of attacks are easily get noticed compared to the other two types of attacks [59].

### 4.2. Network-Probing Attacks

A network-probing botnet attack happens when an attacker continuously collects information about the devices in the network or the network itself [60]. The network-probing attack types include IP address scanning, port scanning, and sending a simple service discovery protocol (SSDP) search query [60]. Unlike DDoS attacks, network-probing attacks are harmless to the functionality of the network. However, these attacks often are performed to gather the target’s information before launching other devastating attacks [57]. In SDN, there is no clearly defined approach that can be used for the prevention of probing attacks. However, techniques such as rate limiting, activity modeling, and implementing distributed firewall and access controls have been used [60,61].

### 4.3. Backdoor Vulnerability

A backdoor is the most common vulnerability threat in a network that integrates IoT. IoT manufacturers often put backdoors on devices for remote debugging [62]. Besides, backdoors can be included in an application by the application developer. In addition, they can be a freestanding application of their own, such as the command and control (C&C) interfaces used in the nodes of botnets [56]. Hackers often exploit this hole and penetrate IoT networks to illegally access network resources and initiate further attacks such as DDoS [63]. There are different backdoor attack initiation mechanisms, but the most common are special credentials, hidden functionality, and unintended network activity [63].

### 4.4. Information Stealing

Nowadays, IoT gadgets are being used from simple home appliances to more advanced operations such as monitoring the health status of patients. In the same token, personal information is becoming more vulnerable to theft via compromised IoT devices and resulting in devastating consequences [64]. For instance, in a botnet attack, a botmaster can generate a command so that the bot on the compromised host collects sensitive and confidential information [48]. In a similar procedure to other botnet attacks, first, an attacker identifies vulnerable devices. Then, the botmaster takes control of attacked devices. Finally, attackers organize all infected devices in a network to gather or steal sensitive information from the IoT devices in the network.

### 4.5. Phishing Attacks

Phishing is a type of social engineering and malware-based attack where attackers steal users’ credentials using fraudulent attempts such as sending emails and inserting a malicious piece of code into the IoT network system [55,65]. Therefore, a security mechanism is necessary to identify the threats, flaws, and countermeasures that may result in phishing attacks on IoT devices. As a result, several tools have been developed to detect phishing attacks in IoT networks [66].

Even though phishing attacks have been investigated a lot and several strategies have been developed, the detection and prevention of phishing attacks in IoT networks remain a challenge [66]. Recently, SDN-based phishing detection mechanisms that are based on deep packet inspection (DPI) that operate on SDN switching devices have been proposed [67].

## 5. BotNet Attack Detection Techniques in SDN-Enabled IoT Networks

In SDN-based IoT, malware detection tools are deployed at the SDN controller to mitigate the IoT devices and the overall network infrastructure from possible malware attacks. The orchestration of IoT networks with SDN will improve the detection of cyber attacks in an IoT network. This improvement is mainly due to the control and data planes in an SDN being separate so that the data forwarding is executed independently of logical procedures of networking protocols [68,69]. However, due to the SDN controller being centralized, it is highly prone to botnet attacks [70,71]. So, to mitigate large-scale botnet attacks that may result in an exploding SDN controller, machine learning-based botnet detection techniques have been proposed in literature [70]. Machine learning-based botnet attack detection techniques for SDN-enabled IoT networks can be done using the classical machine learning and deep learning approaches. These approaches are presented in the following subsections.

### 5.1. Classical Machine Learning Methods

In botnet attack detection, several classical machine learning algorithms have been proposed and tested in literature [28,70,72,73,74,75,76]. One of the limitations of these techniques is that they require extensive feature engineering to achieve optimal features. As a result, they fall short in detecting unforeseen botnet attacks [77]. However, classical machine learning-based botnet attack detection has been an active research area in the deterrence of cyber-attacks in SDN-enabled IoT systems [74]. The most common botnet attack in SDN-based IoT networks is generating spikes of huge traffic that hinder the system’s availability for legitimate traffics. Such types of attacks are called DDoS attacks. One of the techniques for mitigating such attacks is devising a reactive strategy that blocks infected IoT devices from the network.

In the detection of new botnet attacks, a reinforcement learning-based technique that has a real-time attack mitigation capability is proposed in [78]. The challenges in developing reinforcement learning models in botnet detection are low sample efficiency and the task of specifying the reward function. In the proposed model the agent prevents an SDN-enabled IoT network by redirecting traffic, dropping suspicious connections, or allowing traffic that was previously blocked [78]. The proposed techniques have shown better performance in terms of identifying and blocking malicious traffics and increasing the reward value. However, the literature lacks rigorous experiments and quantitative performance evaluations.

In protecting SDN-enabled IoT networks from botnet attacks such as DoS and DDoS, one of the most commonly tested classical machine learning algorithms is a random forest (RF) [28,70,76]. Sarica et al. [28] proposed a random forest for DoS and DDoS attack detection and mitigation using automated feature extraction from network flow traffic. The model was trained using a dataset generated from IoT devices to train the Random Forest classifier at the SDN application layer. The dataset contains six different classes which are benign (normal traffic), DoS, DDoS, port scanning, OS fingerprinting, and fuzzing. The reported performance of the model in identifying normal traffic and attacks such as DoS and DDoS is quite high. Accordingly, the normal traffic identification performance of the model in terms of accuracy, precision, recall, and F1 Score was 99.67%, 96.75%, 92.92%, and 94.80%, respectively. Similarly, the precision, recall, and F1-score of the proposed model in identifying the DoS attack were 97.77%, 99.12%, and 98.44%. And, for DDoS attack detection, the model performance is with a precision of 97.29%, recall of 98.58%, and F1-score of 97.93%.

A random forest (RF) based botnet attack detection technique in an SDN-enabled network using only forwarded information has been proposed in [70]. In the proposed framework, network function virtualization (NFV) detects network attacks and makes a service chain of virtual network functions (VNF). In turn, one of the VNFs keeps monitoring traffic and collects the feature set information that will be transferred into an SDN controller to be used for machine learning-based botnet detection. The RF model is trained using the public botnet dataset CTU-13, BoNeSi, and Mirai. The RF is trained using network traffic information feature sets including source/destination address, port numbers, flags, types of services, protocols, and bytes. The proposed RF algorithm has achieved an accuracy of 100% on CTU-13 and Mirai, and 98% on BoNeSi. After the network traffic is analyzed via a random forest machine learning model, the controller defends the overall network in real time by changing the network’s behavior. Similarly, Arman et al. [79] have used an SDN-enabled home gateway to their proposed random forest machine learning model efficiency in detecting volumetric DDoS attacks. The model receiver’s network flow information from the Ryu SDN controller via Open Virtual Switch installed on IoT gateways so that the traffic will be classified as either normal or adversarial. The reported model’s performance in terms of accuracy was 92%. In addition, Nanda et al. [80] proposed a random forest-based DDoS attack detection system in SDN-enabled IoT networks. In the proposed system, the incoming packet header is classified into either a normal packet or an attack using a random forest model deployed in an SDN controller. Then, normal packets are forwarded to the SDN switches for further processing, and if the packet is in an attack the system pops up a notification. The model was tested on an attack dataset collected by the author, and the reported performance in terms of accuracy, precision, recall, and F1-score was 98.7%, 98.7%, 98.7%, and 98.7%. However, the performance of the model decreases in a scenario where the rate of attack packets is lower. The performance of different machine learning-based algorithms on the detection of low-rate DDoS attacks in SDN-enable IoT has also been tested by Cheng et al. [81]. Often, low-rate traffic attacks go undetected until the data plane of the SDN is overwhelmed and even crashed [82]. So, Cheng et al. [81] have compared the low-rate DDoS attack detection performance of different machine learning models when deployed both on the controller and data planes of the SDN. Accordingly, as shown on Table 4 the RF has shown the highest detection performance both on the SDN controller and SDN switches.

An eXtreme Gradient Boosting (XGBoost) machine learning model, another tree-based algorithm that consists of a number of hyper-parameters to be tuned has also been proposed for botnet malware detection in SDN-enabled IoT networks [72]. The model was trained using three datasets CICDDoS2019, NSL-KDD, and CAIDA-DDoS attack 2007. These datasets are subjected to an efficient preprocessing stage to prevent misleading accuracy and overfitting problems. Then, different features with the most descriptive information had been identified which were used to train the XGBoost in classifying whether the network traffic is a DDoS attack or normal [72]. The reported model performance in a real-time SDN environment was above 99.98% in terms of accuracy, precision, recall, and F1-score. Similarly, AdaBoost-based DDoS attack detection in an SDN-based network has been proposed by Swami et al. [83]. Accordingly, the accuracy, precision, recall, and F1-score of the model were 99.99%, 100%, 99.98%, are 99.99%, respectively.

An ensemble of machine learning techniques that combines decision tree and gradient boosting was proposed by Thorat and Kumar [76]. The proposed model was implemented on the SDN controller and has achieved the highest performance of 98.93%, 97.88%, and 97.94% in terms of accuracy, precision, and recall, respectively. Furthermore, the proposed model has the capability of preventing the DDoS attack on the IoT device and avoids the need to mitigate the attack on the cloud server [76]. Similarly, Aslam et al. [73] proposed a variety of support vector machines (SVM) such as Lagrangian SVM, Finite Newton Lagrangian SVM, Smooth SVM, and Finite Newton SVM that have different types of kernels for the DDoS attack detection and mitigation in SDN-enabled IoT networks. The SVM classifiers were trained using four types of traffic features, that are, rate of source IP (RSIP), a standard deviation of flow packets (SDFP), a standard deviation of flow bytes (SDFB), rate of flow entries on switch (RFES), and the ratio of pair-flow entries on switch (RPFES). According to Aslam et al., [73], these features contain highly descriptive information for distinguishing normal and malicious traffics in SDN-enabled IoT networks. For instance, attack packets have smaller sizes compared to normal packets. In addition, the packet flow rate from normal IoT devices is preset. However, the inter-packet interval of malicious packets is short and random. So, after the proposed SVM models were trained, it was reported that the Finite Newton Support Vector Machine (NSVM) has the highest performance with an F1-score of 94%.

An ensemble of several classical machine learning algorithms has been proposed by Aslam et al. [74]. The proposed framework mitigates the SDN-enabled IoT networks from possible DDoS attacks. The framework is adaptive and multi-layered that the first layer contains ensembles of several classical machine learning algorithms including support vector machine (SVM), Naive Bayes (NB), Random Forest (RF), K-Nearest Neighbor (KNN), and Logistic Regression (LR) classifiers. The model is trained using the IoT’s network. The outputs of the classifiers of the first layer are subjected to voting in which the output from the majority of the classifier wins. This is done in the second layer through ensemble voting (EV). The third layer in the framework measures the real-time live network traffic to detect DDoS attacks. Accordingly, authors tested their proposed framework and the reported result indicates that the framework has better efficiency in four performance metrics of accuracy, precision, recall, and F1-score were 99%, 98%, 96%, and 95%, respectively. However, the proposed model did not take into consideration the DDoS attacks mitigation technique for the botnet attacks on the SDN controller itself. Similarly, Tsogbaatar et al. [75] proposed a model that integrates deep autoencoder as a feature extractor and an ensembled probabilistic neural network (PNN) for anomaly detection, including N-BaIoT and botnets simulated on BoNeSi datasets. The proposed model was tested on a real-time SDN-enabled IoT network testbed, and the model has demonstrated a detection rate in terms of F1-score of more than 99.8% both on benchmark and real-time testbed datasets.

A summary of the literature reviewed in this section is given Table 4. As shown in the table, the performance of these models is quite high once a descriptive feature that contains important information about the network flow has been identified using extensive feature engineering. However, they have a limitation in detecting unforeseen botnet attacks.

**Table 4 sensors-22-09837-t004:** Summary of BotNet attack Detection using Classical Machine Learning in an SDN-enabled IoT Networks.

Author	Method	Dataset	Acc (%)	P (%)	R (%)	F1 (%)
Sarica et al. [28]	RF	Collected by authors [84]	99.67-Normal	96.75-Normal	92.92-Normal	94.8-Normal
			99.67-DoS	96.75-DoS	92.92-DoS	94.8-DoS
			-	97.29-DDoS	98.58-DDoS	97.93-DDoS
Bhunia et al. [68]	non-linear SVM	simulated dataset	-	98	97	-
Park et al. [70]	RF	CTU-13 Dataset	100	-	-	-
		Mirai	100	-	-	-
		BoNesi	98	-	-	-
Alamri et al. [72]	XGBoost	CICDDoS2019, NSL-KDD, and CAIDA-DDoS	99.9	99.98	99.99	99.98
Aslam et al. [73]	Finite Newton Support Vector Machine (NSVM)	Simulated dataset	-	-	-	94
Aslam et al. [74]	Ensemble	Simulated dataset	99	98	96	95
Tsogbaatar et al. [75]	PNN	collected real-time dataset	-	-	-	99.8
		NBaIoT	-	-	-	99.9
Thorat and Kumar [76]	RF+XGBoost	Not explicitly indicated	98.93	97.88	97.94	-
Arman et al. [79]	RF	Testbed data	92	-	-	-
Nanda et al. [80]	RF	Simulated dataset	98.7	98.7	98.7	98.7
Cheng et al. [81]	SVM (controller)	Real-time collected	97	96	97	97
	SVM (switch)	Real-time collected	90	94	95	94
	NB (controller)	Real-time collected	79	72	84	77
	NB (switch)	Real-time collected	66	92	67	77
Cheng et al. [81]	KNN (controller)	Real-time collected	97	96	97	97
	KNN (switch)	Real-time collected	89	93	95	94
	RF (controller)	Real-time collected	97	97	97	97
	RF (switch)	Real-time collected	91	95	94	94
Swami et al. [83]	Adaboost	Simulated	99.99	99.98	100	99.99
Wani and Revathi [85]	Multi-layer perceptron	Simulated	-	98.74	96.43	-
Wang et al. [86]	Dynamic generative self-organizing map (DGSOM)	ISCX-IDS2012	95.41	-	-	-
Zeleke et al. [87]	RF	CICIDS2017	>99.96	>99.51	>99.51	>99.51

### 5.2. Deep Learning Methods

As discussed in Section 5.1, the classical machine learning based botnet detection techniques have proven their effectiveness in botnet attack detection in SDN-enabled IoT networks. But, timely detection, real-time monitoring, and adaptability to new attacks are still serious challenges that should be addressed. This is mainly because classical machine learning techniques use signatures of the already known malware both in training and after deployment. However, some conventional machine learning classifiers such as random forests showed comparable performance to the deep learning classifiers [88]. Nevertheless, deep learning-based techniques including deep reinforcement learning methods have shown effectiveness in real-time monitoring of unseen botnet attacks in SDN-based IoT networks [34,89]. In this sub-section, we will evaluate the role of deep learning techniques in bot detection based on how each has been used. Then, each technique will be reviewed in two ways: first, a short and brief discussion of the technique is presented, and second, a brief look at how the technique has been used in botnet detection is carried out.

#### 5.2.1. Deep Neural Networks (DNN)

Deep learning models are based on the structure of a feed-forward neural network and consist of the input layer, hidden layers, and output layer. A general architecture of DNN is shown in Figure 2. The input layer receives the vector representation, that is, $X=\{x_{1},x_{2},...,x_{d}\}$ of the task to be classified. The feature inference is done through hidden layers, and in the output layer the class vector, that is, $Y=\{y_{1},y_{2},...,y_{d}\}$ associated with the type of data traffic generated in the network. In the DNN with *M* total number of layers and $k_{m}$ being the number of processing units in the mth layer with m=1,...,M and the (m,n)th unit in architecture represent the nth unit in the mth layer, the corresponding input-output of the unit is mapped as [90,91]:(1)$$f_{m,n}\left(w_{m,n},\varphi_{m-1}\right)=A\left(w_{m,n}^{T}\varphi_{m-1}+b_{m,n}\right)$$
where *A* is an activation function such as sigmoid, tanh, and ReLu, $\varphi_{m-1}\in R_{m-1}^{k}$ denotes the output from (m−1)th layer and input to the mth layer, $w_{m,n}\in R_{m-1}^{k}$ denotes the weight vector associated with the (m,n)th unit, and $b_{m,n}$ denotes the bias vector associated with the (m,n)th unit.

The DNN’s hidden units at a particular layer *m*, can be mapped by:(2)$$F_{m}\left(W_{m},\phi_{l-1}\right)={\left[f_{m,1}\left(w_{m,1},\varphi_{m-1}\right),...,f_{m,k_{m}}\left(w_{m,k_{m}},\varphi_{m-1}\right)\right]}^{T}$$
where $W_{m}=\left[w_{m,1},...,w_{m,k_{m}}\right]\in R_{m-1}^{k}\;\times \;k_{m}$ being the mth layer weight matrix. Then, the training is performed by minimizing the loss function given in Equation (Equation 3).
(3)$$J\left(W,x_{i}\right)=E\left(F\left(W,x_{i}\right),y_{i}\right)$$
where the error function E(.,.) is differentiable with respect to W.

Recently, with the advancement of the computing power of devices, the deep neural network has been drawing the attention of many researchers in the field of computer vision, pattern recognition, audio, and text analysis [92]. Similarly, DNNs have been used for detecting botnet malware in SDN-enabled IoT networks [93,94,95]. Khan et al. [93] evaluated different hybridized DNNs with other deep learning models for detecting well-known malware botnet, N_BaIoT. DNN-DNN hybrid model is one of the proposed models and has achieved a botnet detection accuracy of 99.93%, a precision of 99.87%, a recall of 99.86%, and an F1-score of 99.86%. The model performance is better when compared to other hybrid models but with higher time complexity [93].

A simple DNN with three hidden layers containing 12, 6, and 3 neurons with 6 input and 2 output dimensions was proposed by T. A. Tang et al. [95] to detect denial of service (DoS) and network-probing attacks. The proposed model was trained using the NSL-KDD dataset. The technique’s reported performance was with an accuracy of 75.75%, a precision of 83%, a recall of 75%, and an F1-score of 74%. Similarly, Narayanadoss et al. [96] proposed a simple DNN architecture to detect DDoS attacks in vehicular SDN-enabled IoT networks. The DNN architecture is so simple that it has 2 hidden layers having 25 neurons each. The input layer has 50 neurons, and a single neuron on the output layer tells whether the network is under attack. To generate the model training and testing dataset, Narayanadoss et al. [96] used iperf3 to generate the normal and abnormal traffics. The model performance is tested under different scenarios, including varying numbers and speeds of vehicles. The model performance for thirty-five (35) vehicles in terms of accuracy is around 85%, and the precision, recall, and F1-score are around 87%.

A DNN model that detects and classifies 12 different types of botnet attack traffics on a variety of IoT devices in an SDN has been proposed in [97]. The proposed DNN model has three hidden layers with 134, 60, and 26 neurons each. The model was trained and tested using CICDDoS2019 and TON_IoT datasets. The reported classification performance of the model on the CICDDoD2019 dataset in terms of average accuracy, precision, recall, and F1-score, were 93.88%, 68%, 63%, and 58%, respectively. On the TON_IoT dataset, the reported performance of the proposed model in terms of average accuracy, precision, recall, and F1-score, were 98.93%, 93%, 93%, and 95%, respectively [97]. However, the model performance is higher in classifying the data traffic into binary classes.

To overcome the challenges, such as the need for a large labeled dataset and the difficulty of detecting new attack types, Ravi et al. [98] proposed a semi-supervised learning technique that integrates DNN and K-means. The hybridization of DNN and K-means in the proposed model increases the model generalization capability and improves the model detection accuracy in case of unknown attacks. The proposed model is trained on the NSL-KDD dataset, and the reported attack detection accuracy was 99.78% and the F1-score of 99.72%.

Al-Abassi et al. [94] has also proposed an ensemble of DNNs with a decision tree (DT) to detect malware in an industrial control system. The classification output from this *N* DNN model was concatenated via a super vector using a fusion activation function. Then, the concatenated vector is passed on to a DT classifier to detect attacks from the newly merged representation. The proposed model was trained and tested using industrial control system (ICS) datasets [94] and the reported results indicate the ensembles of DNN with DT have shown a better malware detection performance compared to other classifiers like RF, DNN, and AdaBoost. Accordingly, the accuracy, precision, recall, and F1-score of the system were 99.67%, 97%, 99%, and 99%, respectively. However, given cascaded and stacked machine learning models, it is difficult to conclude the system’s performance in real-time. Furthermore, the computational complexity of the proposed model did not present.

A summary of DNN-based botnet detection in SDN-enabled IoT networks is shown in Table 5. The reported results indicate that the ensembles of DNN with other machine learning algorithms improve the detection and mitigation of cyber-attacks in SDN-enabled IoT networks.

#### 5.2.2. Convolutional Neural Networks (CNN)

Convolutional Neural Networks (CNNs) are a class of artificial neural networks designed to automatically and adaptively learn spatial hierarchies of features through back-propagation. CNN has multiple building blocks, including convolution layers, pooling layers, and fully connected layers [91]. The convolution and pooling layers perform feature extraction, and the fully connected (FC) layers map the extracted features into the output traffic classes. As shown in the Figure 3, the input data is convolved by the kernel resulting in a set of linear activated feature maps. Then, each linear activated feature map passes through non-linear activations such as rectified linear units (ReLu). A further modification is done through pooling functions such as max-pooling to downsample the spatial size of the feature map. As the output of the predecessor convolution layer is given as an input to the successor convolution layer, the extracted features become more complex and hold the most descriptive information of the input data [91]. A complete CNN architecture shown in Figure 4 contains a stack of several building blocks depicted in Figure 3.

Training of a convolutional neural network is performed by finding the optimum value of kernels and weights in the convolutional layers and the fully connected layers. The optimum values are those values of the filters and weights that minimize the difference between the model’s output and the ground truth on a dataset. The model performance is calculated using kernels and weights values using a loss function such as cross-entropy. The error is back-propagated through an optimization algorithm such as gradient descent [91].

Deep 2D CNNs have shown tremendous success in computer vision applications. Besides, depending on the dataset types found in several applications, single-dimensional (1D) CNN architectures have been proposed in literature [103]. Assis et al. [104], proposed a 1D CNN model to detect and mitigate DDoS attacks in SDN-enabled IoT networks. The proposed CNN architecture takes IP flow traffic data of a second length in the form of a 3D tensor. The proposed architecture is composed of a stack of two Conv1D and MaxPooling1D layers that are followed by the flatten and dropout layers. Then, the fully-connected layer performs a global model classification. Finally, data is classified into normal or DDoS attacks on the output layer. The proposed model is trained and tested using a dataset simulated by the author and CICDDoS 2019 datasets. Accordingly, the proposed CNN achieved an average detection performance of 99.9% on the simulated dataset with all performance metrics of accuracy, precision, recall, and F1-score. However, the number of hosts used for generating this simulated dataset was two hundred (200) connected with six (6) switches. So, the simulated SDN environment may not suffice to efficiently test the capability of the proposed CNN model against DDoS attacks. This is noticed by the slight reduction in the model’s performance when trained and tested using the CICDDoS 2019 dataset. In this scenario, the reported performance results in terms of accuracy, precision, recall, and F1-score were 95.4%, 93.3%, 92.4%, and 92.8%, respectively. However, the reported results in both scenarios indicate the superiority of the proposed method compared to other machine learning techniques used for comparison. In the same token, Ferrag et al. [97] have shown the superiority of convolutional neural networks in different attack detection compared to DNNs and RNNs. The performance of the CNN model in classifying 12 multi-class attacks using the CICDDoS2019 dataset was 95.12% of accuracy, 91% of precision, 90% of recall, and 89% of precision. Whereas, for the TON_IoT dataset the reported performance of the proposed CNN model in terms of average accuracy was 99.92% [97]. The superior performance is mainly because of its ability for extracting descriptive information from a stream of network packets [105].

Convolutional neural networks have been used to carry out a feature extraction process in hybridized deep learning-driven botnet malware detection algorithms [106]. As we discussed earlier, the convolutional and pooling layers of CNN capture spatial features efficiently. However, due to a lack of temporal information, it is not easy to identify the inter-dependency of features in 2D-CNN. To compensate for this, Liaqat et al. [106] proposed a hybridized deep learning architecture that has cuDNNLSTM layers after the CNN layers. The proposed hybrid model was trained and tested on the Bot-IoT dataset, and the reported results with accuracy of 99.99%, a precision of 99.83%, recall of 99.33%, and F1-score of 99.33%, indicates the superiority of the CNN-cuDNNLSTM architecture in detecting the botnet malware in SDN-enabled IoT networks.

Ullah et al. [107] proposed a hybrid architecture that hybridized CNN and LSTM. Again, CNN was used as a feature extractor where its pooling layer outputs are given to the LSTM model. The proposed framework was trained and tested on CIDDS-01, and the reported performance results in terms of accuracy, precision, recall, and F1-score were 99.92%, 99.85%, 99.94%, and 99.91%, respectively. Similarly, S. Khan et al. [108] have proposed a hybrid CNN-LSTM model for malware detection in an SDN-enabled internet of medical things (IoMT) network. The hybridization of these two models brings together the efficient feature extraction of the CNN and the LSTM’s capability in learning the temporal interdependence of features. The proposed hybrid model has achieved a malware detection performance in terms of accuracy, precision, recall, and F1-score of 99.96%, 96.34%, 99.11%, and 100%, respectively.

In the vehicular environment, where the IoT nodes are mobile, CNN has shown comparably lower performance in the detection of attacks compared to DNN and LSTM models [96]. In such type of mobile environment, there is difficulty in identifying the correct temporal and spatial correlations among the flow of data packets between IoT nodes. The proposed CNN model by Narayanadoss et al. [96] has achieved a 76% attack detection accuracy, while its precision, recall, and F1-score were around 83%. A summary of the literature reviewed in this section is given Table 6.

#### 5.2.3. Recurrent Neural Network

A recurrent neural network (RNN), having a memory capability, learns from the current training input data and prior input information [111]. This behavior is achieved by the feedback connections that allow the network to incorporate the effects of the prior part of the input sequence. So, an RNN model training assumes the dependency of the network output on the previous input sequences. Having this concept into consideration, an RNN has different design patterns. These design patterns include a recurrent connection between hidden units that produce an output at each time step as shown in Figure 5a, a recurrent connection only from the output at a one-time step to the hidden units at the next time step as shown in Figure 5b, and recurrent connections between hidden units that produce a single output for input sequences as shown in Figure 5c [91].

Taking Figure 5a, input sequence *x* is mapped to *o* output values as given in Equation (Equation 4).
(4)$$o^{\left(t\right)}=c+V\tanh\left(b+{\mathrm{Wh}}^{\left(t-1\right)}+{\mathrm{Ux}}^{\left(t\right)}\right)$$
where b and c are bias vectors, U weights in the input-to-hidden connections, V weights in the hidden-to-output connections, and W hidden-to-hidden connections. After post-processing on *o*, for instance, using softmax, a normalized probability of the output $\hat{y}$ is obtained as indicated in Equation (Equation 5). So, learning in RNN is minimizing the difference between the model output vector ${\hat{y}}^{\left(t\right)}$ and the labeled ground truth $y^{\left(t\right)}$, that is, the loss *L*. The learning is achieved via computing the gradient through the network using optimization algorithms such as back-propagation through time. Consecutively, the gradient operation is applied on all parameters of the network [91].
(5)$${\hat{y}}^{\left(t\right)}=softmax\left(o^{\left(t\right)}\right)$$

RNN can be made deeper by stacking additional hidden layers in the network. Deep RNN architectures can perform hierarchical feature learning and have brought a commendable anomaly classification performance even in highly unbalanced training data [112]. However, recurrent neural networks suffer from the vanishing and exploding gradient problem that poses a challenge in the learning process [91,111]. So, new variants of gated RNN, such as long short-term memory (LSTM) and gated recurrent unit (GRU) have been proposed [91].

An LSTM is the most commonly used variant of RNN. In an LSTM, the intuition is incorporating a gated unit that helps to avoid vanishing and exploding gradient problems by controlling the flow of information through the network units. This behavior is achieved through input modulation and forget gets of the LSTM memory cell. So, by introducing context-conditioned self-loops, gradients can flow long in the network. On the other hand, Gated Recurrent Unit (GRU) has a similar concept to LSTM but a single gating unit, that is, forget gate controls the forgetting factor and decision to update state unit [91].

In the previous sections, we have seen the capability of the LSTM model in learning temporal interdependence of features from the stream of data packets in SDN-enabled IoT networks when it is used together with other deep learning models such as CNN. A standalone LSTM has shown its effectiveness in detecting malware in SDN-enabled IoT networks [113]. Hasan et al. [113] has employed an LSTM model that can be integrated into commercially available SDN controllers such as Floodlight, POX, and OpenDaylight. The model was trained, tested, and validated with the state-of-the-art N_BaIoT 2018 dataset. With its inherent nature of remembering the input values at different time intervals, the malware detection performance of the proposed model in terms of average accuracy, precision, recall, and F1-score were 99.96%, 99.93%, 99.88%, and 99.88%.

An LSTM has often been hybridized with different deep-learning models to improve the detection rate of malware attacks in SDN-enabled IoT. For instance, Khan et al. [93] evaluated the LSTM-DNN hybrid model in detecting the well-known malware botnet, N_BaIoT. The model has achieved a botnet detection accuracy of 99.94%, a precision of 99.91%, a recall of 99.86%, and an F1-score of 99.86%. The model performance is comparable to other hybrid models but with the lowest time complexity [93]. Similarly, Javeed et al. [114] proposed an SDN-enabled hybrid deep learning model to detect threats on IoT devices using the publicly available dataset named CICIDS2018. The authors used six threat classes, including benign, DDoS, Bot, and Brute Force to analyze the performance of their proposed model. For threat detection, the combination of Cu-DNNGRU and Cu-BLSTM classifiers is designed. The model has achieved an accuracy of 99.87%, a recall value of 99.96%, a precision value of 99.96%, and a 99.96% of F1 score. Another cyber-attack detection technique by adopting a deep learning-based SDN-enabled model on a fog-to-IoT environment was proposed by Ullah et al. [107]. The authors used two classes of attacks out of the ten classes available from the publicly available dataset, that is, CIDDS-001 used to detect attacks on IoT devices. A hybrid of LSTM and CNN algorithms is trained to detect newly evolving threats in fog-to-IoT environments. With the use of a centralized controller, the proposed model achieves a better result when compared to the previous state-of-the-art architectures. The LSTM-CNN model detects the attaches with high accuracy and precision of 99.92% and 99.85%, respectively. The well-known deep learning algorithms such as RNN, LSTM, and Gated Recurrent Unit (GRU) are proposed by Alshra’a et al. [115] to identify Denial of Service (DOS) attack on Software Defined Networks (SDN). The authors used 48 features out of 83 based on literature recommendations in the SDN dataset to train the models. From the three proposed architectures, LSTM shows the finest result by accuracy and precision to detect Botnet, web-attack, brute force attacks, and exploitation User-to-Root attacks.

A summary of the literature reviewed in this section is given Table 7. As shown in the table, RNN and its variants have been used to detect botnet attacks in SDN-enabled IoT networks due to their capability of learning the temporal interdependence of features from a stream of data packets.

#### 5.2.4. Deep Auto-Encoder

Deep auto-encoder are constructed by adding more hidden layers both in the encoder and decoder sides of the deep belief network as shown in Figure 6. The input to the encoder is raw data or input feature, and an output from the decoder is a reconstructed approximation of the original input data. An auto-encoder is an unsupervised learning algorithm that uses back-propagation to approximate the input by minimizing the reconstruction error. The encoder allows the transformation of the original input data, $x_{i}$, into latent representation $h(x_{i})$, and the decoder reconstructs the approximated version of the original input data $\hat{x_{i}}$ from the latent representations. Here, the error measures the difference between $x_{i}$ and $\hat{x_{i}}$, usually defined as using mean square error or cross-entropy [120].

There are different types of auto-encoders, including sparse, denoising, variational, and convolutional auto-encoders. The deep auto-encoders are the most commonly used to efficiently extract descriptive information about the botnet malware from packet level, flow level, and system metrics data [121,122,123]. In line with this, Tsogbaatar et al. [121] proposed a framework that dynamically deters malware attacks in an SDN-enabled IoT network. The framework contains anomaly detection, intelligent flow management, and device status forecasting modules. In the process of detecting dynamic attacks, the extraction of descriptive features was carried out using the integrated deep encoder and deep probabilistic neural network, that substituted the deep decoder part of the deep auto-encoder architecture. The proposed framework was trained and tested on N-BaIoT benchmark and simulated datasets. The reported model performance on these datasets with different data imbalances is astonishing, with an accuracy of 99.8% and 99.9%, and an F1-score of 99.95% and 99.47% on simulated and N-BaIoT datasets, respectively.

The other difficult phenomena in the SDN-enabled IoT network that poses a challenge for machine learning-based anomaly detection is the time needed for the SDN controller to acquire network flows and processing feature engineering tasks [124]. Accordingly, Ujjan et al. [122] proposed a stacked auto-encoder (SAE) deep learning model that detects DDoS traffic with the help of sFlow and using adaptive traffic sampling at a data plane of the SDN. Then, the SAE model implemented in the SDN control plane uses the information obtained from the data plane to determine whether the traffic is malicious or normal. The proposed model was trained and tested using a real-time testbed network flow dataset obtained with sFlow and adaptive traffic sampling. The reported performance of the model shows that the sFlow-based classification achieved an accuracy of 91%, a precision of 95%, a recall of 83%, and an F1-score of 88.10%. However, symmetrically stacked auto-encoders require high computational and memory resources as the number of stacks increases [123,125]. This complexity poses a challenge for real-time malware detection in SDN-enabled IoT networks. To address this challenge and provide real-time protection for an SDN-enable IoT network, Krishnan et al. [123] proposed a framework based on a non-symmetric stacked auto-encoder for reducing the input feature set. Then, the reduced feature set is used as an input for the RF module where the traffic is classified as malicious or not. Generally, the proposed framework combined multiple behavioral analysis and anomaly detection techniques to intercept and detect malware attacks at multiple stages and planes of an SDN. The model was trained and tested using NSL-KDD and CICIDS2017 datasets. Accordingly, the reported performance of the model in terms of average accuracy, precision, recall, and F1-score was 99.3%, 99.8%, 99.5%, and 99.4%, respectively. Furthermore, the proposed model saved the training time by 94.96% and memory usage by 90.8% compared to symmetric SAE with improved performance.

A summary of the literature reviewed in this section is given in Table 8. As shown in the table, deep auto-encoders have been an efficient feature extractor, and when used with other classifier models they result in higher classification performance.

## 6. Discussion

The orchestration of IoT networks with SDN has greatly improved IoT devices’ susceptibility to cyber-attacks. This immunity is mainly because an SDN decouples the control and data plane of the traditional network architectures and enhances IoT networks’ interoperability, scalability, resource usage, control and management, and reconfigurability. Furthermore, SDN can provide sufficient computational resources for deploying complex anti-malware applications. However, SDN-enabled IoT networks are still vulnerable to botnet attacks. In literature, classical machine learning and deep learning-based techniques have been proposed to alleviate these attacks. So far, reviews undertaken in this area focus on botnet malware detection on either IoT or SDN networks separately. However, SDN-IoT orchestration has brought challenges from security and other aspects [128]. As a result, several ML/DL-based solutions have been proposed in the literature, and it is an active research area. These claims can be observed in the rise of literature published in this area since 2020 as shown in Figure 7. So, this review focused on ML/DL-based botnet attack detection techniques provided for SDN-IoT orchestrated networks.

Classical machine learning-based botnet detection techniques have proven their effectiveness in botnet attack detection as shown by their performance in Table 4. However, these techniques are highly dependent on extensive feature engineering. As a result, they use signatures of the already known malware both in training and after deployment. This hinders their performance in case of unknown botnets. So, the number of literature that proposes classical machine learning techniques for botnet detection in an SDN-IoT network is small as shown in the pie chart of Figure 8. Figure 9 summarizes the number of papers (in pink) as well as the percentage weight (in green) of classical machine learning techniques reviewed in our work. Among classical machine learning techniques proposed in the literature, random forests have been used in most of the literature as shown in Figure 9. Random forest is one of the few machine learning algorithms that perform well in both classification and prediction tasks with higher performance. Furthermore, it performs well in case of missing a portion of data features and is less prone to overfitting.

On the other hand, deep learning techniques such as deep neural networks, convolutional neural networks, long-short term memory, recurrent neural networks, and deep auto-encoders are being used in botnet attack detection in SDN-enabled IoT networks. As shown in Figure 10, among the reviewed literature, DNN-based ones take the highest share. In papers such as [94,98] DNN based automatic feature extractor coupled with classical machine learning techniques based classifiers such as decision tree (DT) and k-means in classifying the data traffic.

Generally, deep learning techniques have shown their effectiveness in real-time monitoring botnet attacks in SDN-based IoT networks. This is mainly because DL models do not require extensive feature engineering and have higher capacity compared to classical ML techniques. This can be observed in Table 5, Table 6, Table 7 and Table 8. In addition, deep learning techniques have shown their efficiency in detecting previously unseen botnet attacks in real-time [107,122,123]. Furthermore, the capability of recurrent deep learning models such as RNN, LSTM, and GRU in learning temporal interdependence of botnet attack features from the stream of packets plays a key role in detecting unseen attacks [113,114]. Besides, these recurrent deep learning algorithms have been hybridized with other deep learning techniques such as CNN and DNN to improve the botnet attack detection performance [93,107,108].

However, hybridizing classical ML classifiers with deep learning techniques such as deep autoencoders is becoming an alternative solution [121,123].

## 7. Open Challenges and Future Direction

The SDN-IoT orchestration has improved IoT networks’ resilience against attacks. Also, the reported performance of proposed ML methods shows the capability of ML techniques in detecting botnets. However, the existing SDN controllers lack standard attack detection security mechanisms [19] that need to be standardized and enhanced. Apart from the issues in an SDN controller, there are challenges to be addressed to have a robust botnet attack detection ML technique that can work in real-time.

First, limited benchmark datasets exist for training and testing the proposed ML models. In addition, due to the continuous change in the signature of botnet attacks, ML models trained on a given attack signature might not be successful in detecting a previously unseen attack. To alleviate these challenges, archiving a benchmark dataset is an important step. Besides, the ML techniques should incorporate human-expert in the loop to maintain continuous learning.

Second, to obtain an ML-based attack detection mechanism that can work in real-time, the computational complexity associated with these models should be as minimum as possible. In addition, the ML models did not provide the reason behind their decision to classify the stream of packets as normal or malicious. To mitigate these challenges, ML models should be lightweight, and the ML techniques should be designed in a way to provide a reason behind classifying a given stream of packet traffics into an attack or a normal.

Finally, we recommend the collaboration of ML and cyber-security scientists in the development cycles of robust ML techniques that can continuously learn and prevent unseen attack signatures in order to achieve a secured and seamless operation in SDN-IoT systems.

## 8. Conclusions

Automating botnet attack detection in SDN-enabled IoT networks is an important step for the optimum detection of both known and unseen attacks on a real-time basis. To this end, several works of literature that proposed machine learning techniques have been published. So, in this paper, we investigated those research studies that applied both classical machine learning and deep learning techniques for detecting botnet attacks in SDN-enabled IoT networks. To achieve the goals of this review, we identified research questions to be addressed. Furthermore, to identify the literature to be surveyed, literature search strings, publication databases to be searched, and inclusion and exclusion criteria have been determined. Then, botnet attack detection techniques for SDN-enabled IoT networks are structurally categorized and summarized to give an insight to the reader on the dataset used, the advantage and limitations of both ML and DL techniques, and the reported performances of the proposed model in the literature.

Malicious actors often take advantage of vulnerabilities on IoT devices to inject bots and get a foothold in SDN-enabled IoT systems. Once they access these IoT gadgets, they can generate devastating attacks through or on these compromised IoT devices. So, devising robust ML-based botnet detection that can operate in real-time is imperative in SDN-enabled IoT systems. In this regard, the hybridizing machine learning techniques have shown the potential for detecting botnets with known and unseen signatures. However, a more rigorous evaluation of these models on different network settings is required before relying on them. As a result, human experts should be incorporated into the deployment loop of these models to have the capability of continuous learning and discovering unseen botnet signatures.

In addition, proposed ML techniques in the literature have to be evaluated in terms of their computational complexity to realize an ML model that can execute in real-time. However, as indicated in Table 5, Table 6, Table 7 and Table 8 apart from metrics used to measure model performances, proposed techniques are not evaluated in terms of their computational complexity.

Besides, well-organized real datasets which continuously updated to incorporate newly detected botnet signatures should be used to train and test ML techniques proposed by different authors. These enable benchmarking and making an objective evaluation of proposed ML techniques.

In conclusion, different works of literature have shown the potential of ML techniques to detect botnet attacks in SDN-orchestrated IoT networks. This potential of ML techniques in building a robust botnet detection system that can operate in real time can be harnessed by addressing the existing challenges. 

## Figures and Tables

**Figure 1 sensors-22-09837-f001:**
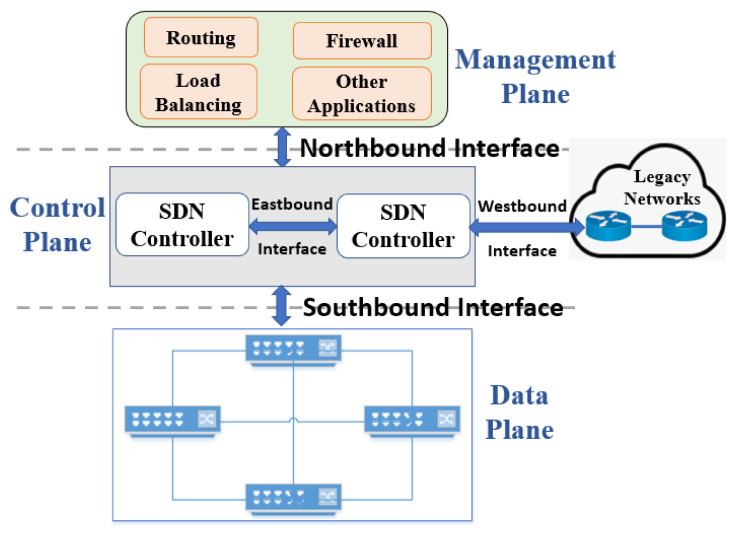
Architecture of an SDN system [19].

**Figure 2 sensors-22-09837-f002:**
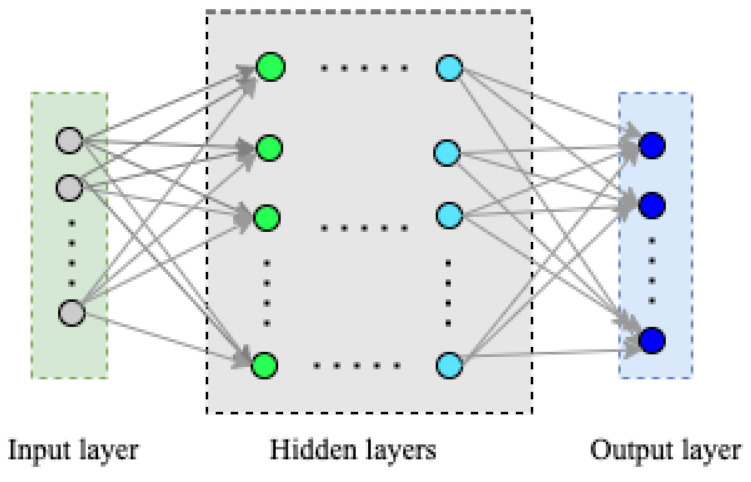
A general architecture of Deep Neural Network.

**Figure 3 sensors-22-09837-f003:**
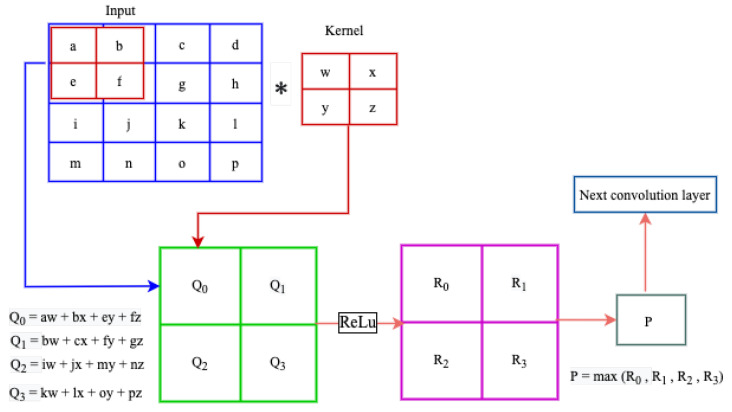
Components of a typical CNN layer.

**Figure 4 sensors-22-09837-f004:**
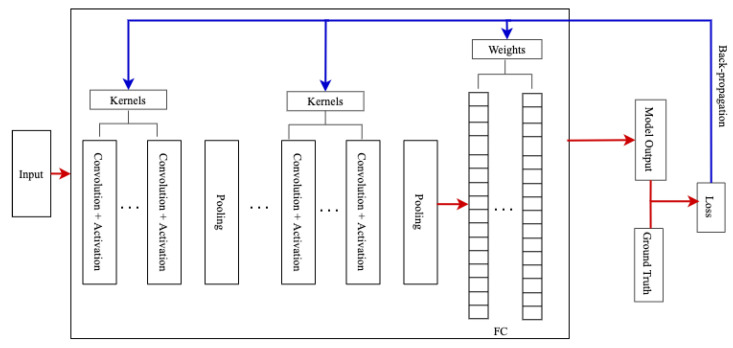
A typical CNN Architecture.

**Figure 5 sensors-22-09837-f005:**
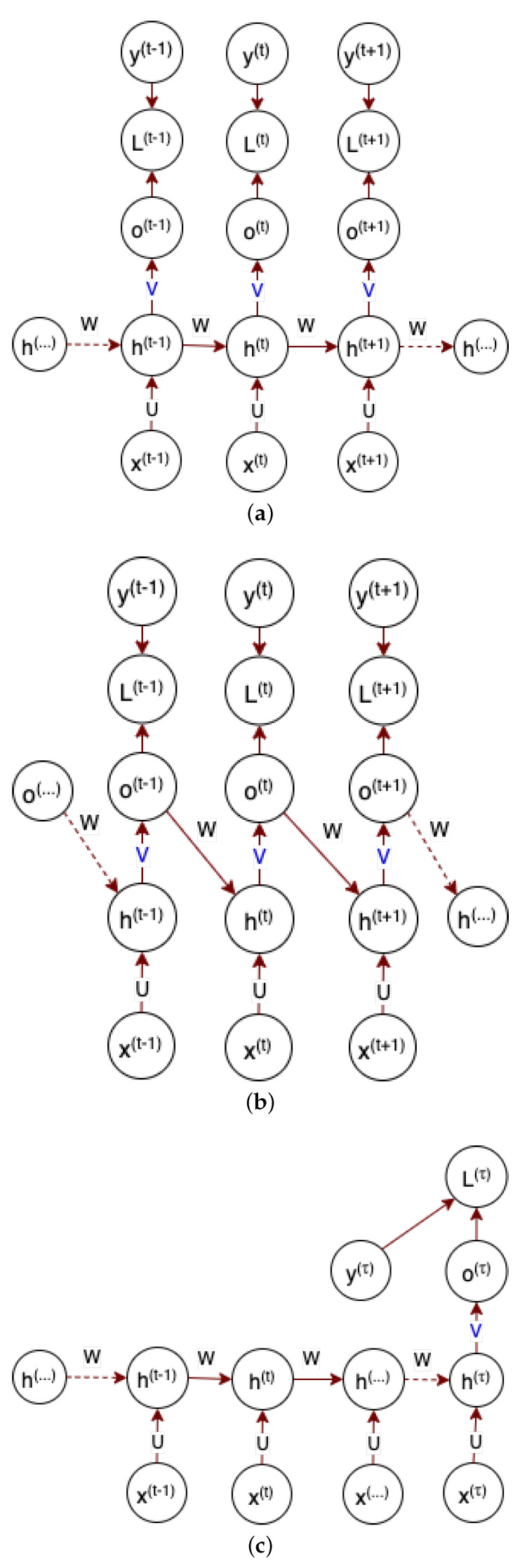
Recurrent neural network possible architectural patterns.

**Figure 6 sensors-22-09837-f006:**
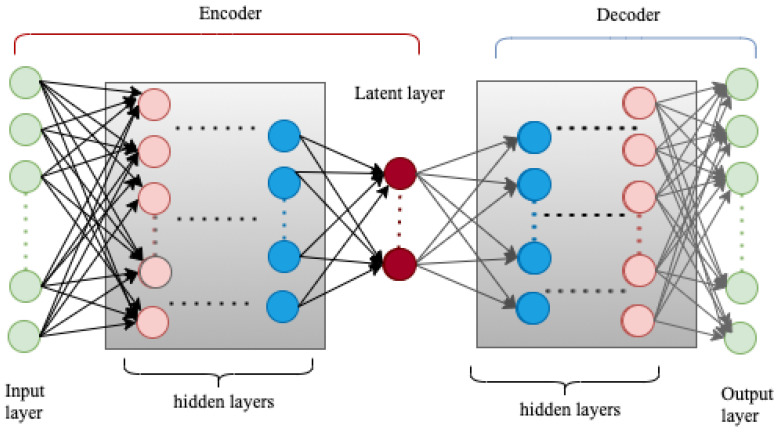
A Deep Autoencoder Architecture.

**Figure 7 sensors-22-09837-f007:**
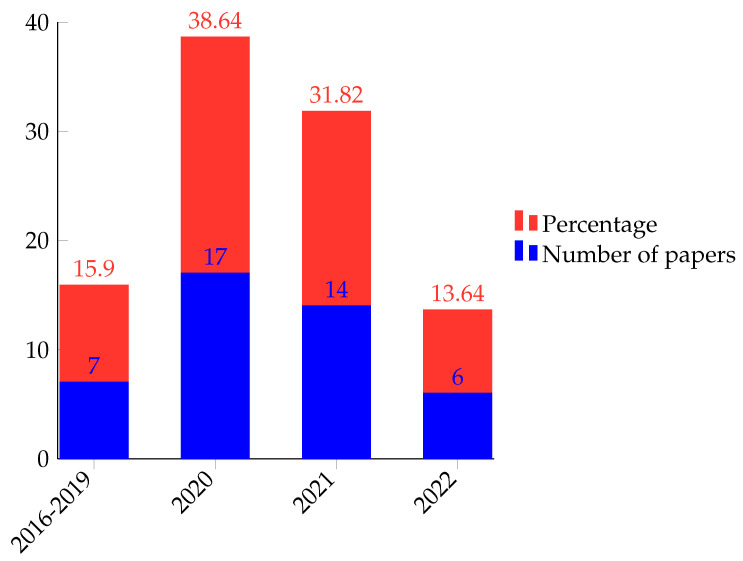
Yearly distribution of reviewed research papers.

**Figure 8 sensors-22-09837-f008:**
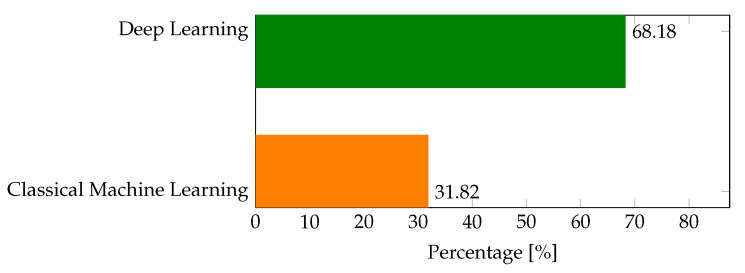
Paper distribution in terms of classical machine learning and deep learning-based techniques.

**Figure 9 sensors-22-09837-f009:**
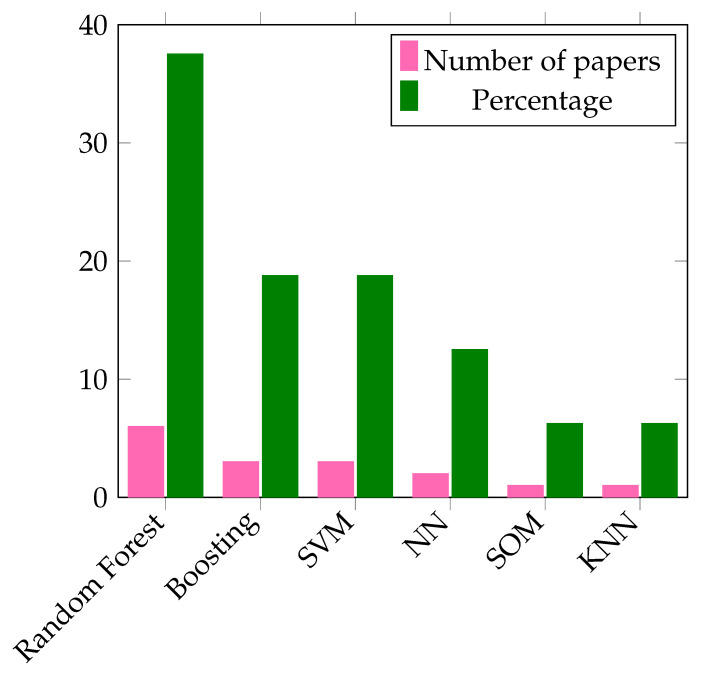
Distribution of reviewed classical machine learning techniques.

**Figure 10 sensors-22-09837-f010:**
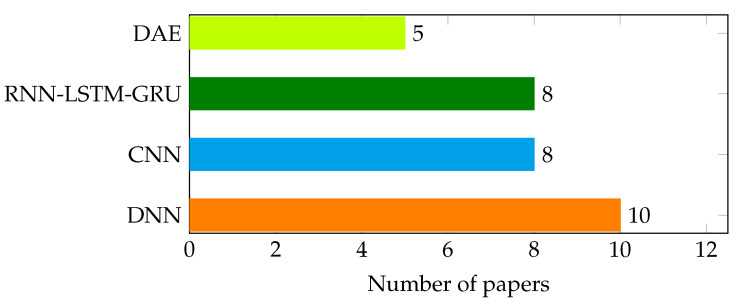
Distribution of reviewed deep learning techniques.

**Table 1 sensors-22-09837-t001:** Summary of Related Works.

Author and Citation	Contribution	Limitation
Pajila and Julie [47]	Reviewed the potential of using machine learning techniques as one DDoS attacks detection mechanism in SDN networks	Few works of literature were included in their work
Shinan et al. [48]	Surveyed literature that focuses on detecting botnets using machine learning techniques in traditional networks	Did not focus on SDN-enabled IoT networks botnet attacks
Snehi et al. [51]	A detailed survey and discussion on improving the performance of Software-defined cyber-physical systems through architectural redesigning have been presented.	The contribution of machine learning techniques in improving the cyber-physical security of SDN-enable IoT networks did not present.
Cui et al. [52]	Comprehensive review on DDOS identification. Classification of DDOS detection mechanisms is proposed, that makes it	In this survey botnet detection using a machine learning approach on SDN-based IoT devices is not conducted.
Aversano et al. [53]	Summarizes recently conducted studies in the area of deep learning applications on IoT Security. Identifies the datasets used by different deep learning architectures for IoT security.	The review did not specify which type of security threat and machine learning-based solutions for botnets in an SDN-enabled IoT.
Ismael et al. [54]	Comprehensive surveys of DDoS detection and mitigation techniques are made	Recommendation of the selected architecture or technique is not properly addressed. The survey did not include SDN-IoT botnets

**Table 2 sensors-22-09837-t002:** Research questions for the review.

No.	Research Question	Aims to Answer
1	What are the major botnet attacks in SDN-IoT networks?	To investigate the major botnet attacks in SDN-IoT networks
2	What machine learning techniques were used in deterring botnet attacks in SDN-enabled IoT networks?	To identify the commonly used machine learning techniques for preventing botnet attacks in SDN-IoT networks
3	How machine learning techniques were used in deterring botnet attacks in SDN-enabled IoT networks?	To acquaint the proposed machine learning techniques for detecting and mitigating botnet attacks in SDN-IoT networks
4	How successful the proposed machine learning techniques were in deterring botnet attacks in SDN-enabled IoT networks?	To analyze and compare proposed machine learning techniques in detecting and mitigating botnet attacks in SDN-IoT networks

**Table 3 sensors-22-09837-t003:** Inclusion and exclusion criteria for paper selection.

Inclusion Criteria (IC)	Exclusion Criteria (EC)
IC1: The papers are in the field of BotNet attack.	EC1: Papers that are not conducted in SDN-IOT.
IC2: The papers have to study different BotNet attacks on SDN-IoT devices.	EC2: Publications not peer-reviewed, abstract, an editorial letter and book review, scientific report.
IC3: The paper should be published in reputable journals or recognized Conference proceedings.	EC3: MSc and Ph.D. thesis, Posters, Seminar.
IC4: The studies should be written in English.	EC4: Studies that are published prior to 2016.
IC5: Published between 2016 and 2022.	

**Table 5 sensors-22-09837-t005:** Summary of BotNet attack Detection using Deep Neural Network in an SDN-enabled IoT Networks.

Author	Method	Dataset	Acc (%)	P (%)	R (%)	F1 (%)
Khan et al. [93]	DNN-DNN	N_BaIoT	99.93	99.87	99.86	99.86
Al-Abassi et al. [94]	DNN+DT	ICS	99.67	97	99	99
Tang et al. [95]	DNN	NSL-KDD	75.75	83	75	74
Narayanadoss et al. [96]	DNN	Simulated data	85	87	87	87
Ferrag et al. [97]	DNN	CICDDoS2019	93.88	68	63	58
		TON_IoT	98.93	93	93	95
Ravi et al. [98]	DNN+K-means	NSL-KDD	99.78	-	-	99.72
Makuvaza et al. [99]	DNN	CICIDS 2017	96.67	97.21	97.29	97.25
Ravi et al. [100]	Deep ELM	Simulated	97.9	97.2	97.6	97.2
		UNB-ISCX	96.28	95.16	97.27	96.2
Maeda, Shogo et al. [101]	DNN	CTU-13 and ISOT	98.7	98.99	99.70	99.34
Sattar et al. [102]	DNN-LSTM	N_BaIoT	99.99	99.99	99.99	99.99

**Table 6 sensors-22-09837-t006:** Summary of BotNet attack Detection using Convolutional Neural Network in an SDN-enabled IoT Networks.

Author	Method	Dataset	Acc (%)	P (%)	R (%)	F1 (%)
Narayanadoss et al. [96]	CNN	Simulated data	76	83	83	83
Ferrag et al. [97]	CNN	CICDDoS 2019	95.12	91	90	89
		TON_IoT	99.92	-	-	-
Assis et al. [104]	CNN	Simulated data	99.9	99.9	99.9	99.9
		CICDDoS 2019	95.4	93.3	92.4	92.8
Liaqat et al. [106]	CNN-cuDNNLSTM	Bot-IoT	99.99	99.83	99.33	99.33
Ullah et al. [107]	LSTM-CNN	CIDDS-01	99.92	99.85	99.94	99.91
Khan et al. [108]	CNN-LSTM	Not explicitly indicated	99.96	96.34	99.11	100
Haider et al. [109]	CNN	CICIDS-2017	99.45	99.57	99.54	99.51
Wang et al. [110]	CNN	real-time collected	97	97	99	96

**Table 7 sensors-22-09837-t007:** Summary of BotNet attack Detection using RNN, LSTM and GRU in an SDN-enabled IoT Networks.

Author	Method	Dataset	Acc (%)	P (%)	R (%)	F1 (%)
Khan et al. [93]	DNN-LSTM	N_BaIoT 2018	99.94	99.91	99.86	99.86
Ullah et al. [107]	LSTM-CNN	CIDDS-001	99.92	99.85	99.94	99.91
Hasan et al. [113]	LSTM	N_BaIoT 2018	99.96	99.93	99.88	99.88
Javeed et al. [114]	Cu-DNNGRU + Cu-BLSTM	CICIDS2018	99.87	99.87	99.96	99.96
Alshra’a et al. [115]	RNN- 48 feat.	InSDN	98.09	97.89	99.65	98.77
	RNN-6 feat.	InSDN	91.11	89.94	99.70	94.51
Alshra’a et al. [115]	LSTM- 48 feat.	InSDN	98.87	98.84	99.70	99.27
	LSTM-6 feat.	InSDN	92.57	92.13	98.77	95.33
Alshra’a et al. [115]	GRU- 48 feat.	InSDN	98.20	97.94	99.75	98.84
	GRU-6 feat.	InSDN	91.31	90.17	99.54	94.62
Malik et al. [116]	LSTM+CNN	CICIDS2017	98.6	99.37	99.35	99.35
salim et al. [117]	LSTM	testbed	96.1	98.38	93.03	94
Yeom et al. [118]	LSTM	Collected real network flow traffic	92	-	-	-
Fredj et al. [119]	LSTM	Capture the Flag (CtF)	-	-	-	93.35
	RNN	Capture the Flag (CtF)	-	-	-	92.90

**Table 8 sensors-22-09837-t008:** Summary of BotNet attack Detection using Deep Autoencoder in an SDN-enabled IoT Networks.

Author	Method	Dataset	Acc (%)	P (%)	R (%)	F1 (%)
Tsogbaatar et al. [121]	DAE_EPNN	Simulated	99.8	-	-	99.95
		N-BaIoT	99.9	-	-	99.47
Ujjan et al. [122]	SAE	real-time testbed (sFlow)	91	95	83	88.1
		real-time testbed (Adaptive Polling)	89	92	78	85
Krishnan et al. [123]	non-symmetric deep SAE + RF	NSL-KDD and CICIDS2017	99.3	99.8	99.5	99.4
Ahuja et al. [126]	SAE	Mendeley data repository	99.75	99.69	99.94	99.82
Choobdar et al. [127]	SAE	NSL-KDD and CICIDS2017	98.5	-	-	-

## Data Availability

Not applicable.

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
