# Peer review of "Review of Botnet Attack Detection in SDN-Enabled IoT Using Machine Learning"

_sensors, 2022, doi:10.3390/s22249837_

Round 1

Reviewer 1 Report

This paper aims to provide a review of work that has been recently conducted ( 2019-present) on the use of Machine learning within  SDN enabled IoT networks. This is a fairly narrow domain.  The case for this narrow focus is in my view not made strongly enough.  A number of works are reviewed . The summary tables are of value, but the writing is difficult to work though , particularly with the intertwining of results of others work along with the explanations of the ML techniques. Some substantive restructure could benefit the overall readability of the paper.    There are either some misunderstandings as to the risks to IoT from botnets, and the execution of attacks form IOT botnets. This could be down to a poor explanation. See detailed notes following.

Overall the paper is an ambitious piece of work, and there is no doubt that there is value in a review of such literature. As it stands however ti needs better contextualization and an improvement of the flow and communication of this significance to the reader.

Specific Notes

Page 1

Abstract - Fair

line 8 - Remove "so,". Sentence reads fine as "The aim .."

line 9 remove "an", Reads as in SDN-enabled IOT...

line 10 - bee ? 

        In doing so, first, major botnet ->  Initally the first.... "

line 11 "secondly a commonly used... networks has been briefly discussed"

line 12 --- Thirdly - at this point the section should be re-written to let the reader know that there are three points coming.

Focus on, and highlight  the results  obtained in this work. be clear to allow readers to discern vlue of the paper.

line 27 - cite evidence for these claims.

line 31 - place citation [3] directly after outpost report.

Page 2 

line 36 - reword - "An IoT network, is a network of "

line 43 - justify claims of IoT being vulnerable to Botnets- only if not properly secured, both form credentials and architecture perspective.

Reduced storage an inability to run anti-malware, are not reasons for their vulnerability

lines 47-54 - consider the relevance of the differing botnet architectures and how/if this relates to the remainder of the paper. For the purposes it should suffice that botnets exist and are controlled by a third part.

line 55-60 - what is being describes is not unique ot an IoT botnet, but is standard mode of operation

line 59 the Mirai botnet, remove "an"

line 63 malware rather than malwares

line 65 - space before [11] and after "."

line 66 - I disagree with the claims of these (Mirai and BASHLITE) botnets being used for theft of information. You need to make a stronger case. These are primarily used for denial of service! the same applies to botnets generically - largely for spam/DDoS rather than info ex-filtration.

lines 72 --- This is not making a particularly strong case for why SDN  based IoT nets are more secure. How are the specific weaknesses identified in the previous section mitigated. Consider the viability for SDN for these types of deployments - eg wearables ?

line 84 - remove basically

line 86 - up on ? upon ?

Page 3

line 93 - no introduction of northbound/southbound...

line 109 -  is the mention of medical and agriculture relevant here ? These really only serve as self citations for some of the authors. If they are relevant, are these the best examples of papers ?

line 110 - have -> has

line 122-125 - this is not a full sentence, Reword.

Page 4

lines 134-139. You are talking about two different papers here, but it reads that it is a continuation of the discussion around [39] "the survey"

Table 1 - in what order are these presented,  its not alphabetical, or citation order ?

The related work section points to some work, but these to not have a particularly strong tie to SNT/IoT and Ml. Methods and results are not discussed here as to what is appropriate for this work. for a meta-review of literature, there would normally be a an exception of significantly more work covered, though this could be handled by being clear that the related works is primarily other survey research in the field.

Page 5

Table 2 S.No == ??

Alg 1 - Why are  OS fingerprinting  and Fuzzing included in the search terms? You are missing brute forcing attacks  or credential stuffing which are a prime propagation vector of botnets. 

Similarly "Cyber attack on SDN-IoT" seem a little restrictive

line 168 - Malware is a collective term (rather than malwares)

line 171-173 - You list a number of attacks here but this does not align with the criteria used in the search terms . What exactly is a backdoor attack ?  A backdoor can be provisioned on a compromised system, but this is not an attack in its own right ? You omit scanning which you explicitly search for , would this (along with OS fingerprinting) fall under network probing ?

line 175 - be clear if this is the IoT network perpetrating or receiving the attack. There is a important difference.  DDoS tends to be perpetrated by IoT botnets, the targets of the attack are almost always traditional systems/platforms.

Page 6

Alg 2 - these keywords and their relation to the searhc string in alg 1 needs ot be carefully explained as this is the basis on which the review of literature is constructed. as it stands there is some disconnect.

consider re-arranging to bring table 3 closer and before alg 2 ?

Page 7

Is IC6 not a subset of IC3 ?

EC4 - rephrase as prior to 2019 , or published in 2018 or earlier.

line 181 - IRC is almost extinct, and most botnets now use more complex crypto protected C2 frameworks. Be clear that as to what target ti, the target victim, or the target systems of the botherder that will undertake the attack.

line 185 - again need to clarify if IoT is the victim or aggressor 

line 206 - this is a very bold claim, given there is no citation.

Page 8

line 218 - for a bulk infection this doesn't really seem plausible, for a targeted attack definitely, but that is then unlikely to be a botnet.

It is not clearly how the SND and IOT specific elements relate to phishing here. Targeted phishing is a potential case, but the vast majority of phishing activity takes place using compromised web systems elsewhere where they phishing content can be hosted. The other element of the phishing operation is the spam transmission of the phishing links.  Be clear  how the IOT space interacts here. Unlikely to be used for hosting due to constrained environs ( and in some cases compiled in web content), similarly they are typically constrained wrt email /smtp capabilities.

Page 9 

line 275 - Is this were the OS fingerprinting and fuzzing comes from. What would fuzzing be in this context - fussing of inputs or protocols ?

line 298 - just Nanda et al ?

306 - fix grammar "he rate of attack packet lower"

Page 10

line 327 - why initials here ? (and see line 331) This is a departure form earlier citation/attribution style. Be consistent, typically initials are omitted.

Page 11

Table 4 provides a useful summary of the discussion. It is not immediately apparent however as to what the horizontal bold lines delineate.

Page 12

Check citations wrt initials.

Page 14

Table 5 - good summary of work, and a lot more accessible than the text. Note the inconsistency citing Tang and Ravi. both Ravi papers are by the same pair, and both published in 2020, these would be typically 2020a and 2020b , however the numeric citation style makes it quite clear as to which you referrer. There is i only a single Tang et al. paper [84] so it is unclear why the inconsistency with citing this as T.A Tang et al.

Page 21

line 663 - You mention that  the SND-IoT orchestration has brought its own challenges. What are these (or refer to where they are discussed, or alternately to a suitable citation)?

line 666- Fig6 to be moved earlier. 

line 675 - pie chars are almost never the appropriate solution, consider a horizontal histogram which would be much more  readable, and more compact.

Page 22

Fig 6 - it would be preferable to plot the two differing values on opposing axes. Be clear that % is calculated of the total number of papers. Is this really relevant however. a count of papers would be sufficient ? This image could be much smaller.

Page 23

Figure 8 - if you are going to use both item (see notes on figure 6) it would be preferable to stick with the same colour mapping to remove the need to explain in ext as you did in circa line 675-678. A horizontal plot may also make this more compact. consider how the data is presented, and the need for both number and % plots

The conclusion is rather brief. There needs to be a tying together of the concepts covered, and a clear highlighting of the most significant points for a reader to take away form this. How could this material be extended or built on by other researchers ?

References

line 726 - [3] set author correctly.

line 740 - [8] How/where published, incomplete ref

line 778 - [22] this is a chapter and this needs the title of the book it is part of 

line 978 [92] Error in  title : &amp

Author Response

Dear Reviewer, thank you for your professional service and valuable comments. I have uploaded the response for your comments. 

Reviewer 2 Report

1. In this paper, the authors propose a survey on Botnet Attack Detection using   Machine Learning techniques for  SDN-enabled IoT.

2. The paper is well-written and well-structured.  

3. The authors are invited to add a subsection in the introduction which gives a short overview of the SDN notion.

4. A figure describing the structure of SDN will be helpful too.  

5. The authors may also add a graphical illustration of the structure of the paper.  

6. The article contains some English mistakes which need to be corrected.  

7. e.g.,  Line 10:  "SDN-IoT networks have bee thoroughly discussed"  

8. Line 152: "with their major contribution" ==> "with their major contributions"  

9. The authors may include the following references in their study: -

https://www.sciencedirect.com/science/article/abs/pii/S0045790622000337
https://ieeexplore.ieee.org/abstract/document/9626266
https://www.mdpi.com/2079-9292/9/3/413
https://ieeexplore.ieee.org/abstract/document/9834261
https://dl.acm.org/doi/abs/10.1145/3433174.3433614

10. Line 65: "are indicate in algorithm 1 and 2, respectively" ==> indicated ... algorithms  

11. Did the authors implement algorithm 2 manually or automatically?  

12. A section about open challenges and future directions is missing.  

13. The authors need to indicate the sources of the figures they borrowed from external documents.  

14. It is preferable to insert Figures 6, 7, and 8 earlier in the article in Section 3. 

15. The conclusion is relatively too short compared to the article's length.                  

Author Response

Dear reviewer, thank you for your comments and I have uploaded the response for each comment. 

Round 2

Reviewer 1 Report

Round 2 review

The paper has improved notably, both in terms of its readability and the attention to detail and consistency.  The authors have addressed the major issues previously raised in a suitable manner. A few minor typographical and language issues remain as noted below.

line 29 - valuable and/or sensitive

line 85 Ope nFlow -> OpenFlow

line 145 - could this ([25]) just be Wang et al ?

table 4 - splitting a table is never idea, but if it must be done, consider shuffling rows so that Cheng et al [82] is as a single row is not split over pages. Consider widening the dataset column and reducing the % ones this would result in a shorter table, improving readability

Previous points about consistency in numbers raised in the prior review applies here too, would suggest left aligning numbers?

line 540 "known by its memory" <- reword

Table 6/7 - not apparent what ordering/ranking is used here

Figure 7 - while functional as is, this could be more compactly represented with horizontal bars a in Fig 8.

Fig 8 - the exact % is less interesting than the number of papers falling in these categories

Fig 10 - Workable as a pie chart but could be more concisely represented as fig 8 if need be.

line 759 - Tables 5-8

Author Response

Dear reviewer, thank you for your feedback after the first revision. We have highlighted the sections and lines we improved as per your recommendation. 

With regards,

Authors